# Boot and Switch: Alternating Distillation for Zero-Shot Dense Retrieval

**Fan Jiang, Qiongkai Xu, Tom Drummond** and **Trevor Cohn**[*]
School of Computing and Information Systems
The University of Melbourne, Victoria, Australia
`fan.jiang1@student.unimelb.edu.au`
`{qiongkai.xu, tom.drummond, trevor.cohn}@unimelb.edu.au`

## Abstract

Neural 'dense' retrieval models are state of the art for many datasets, however these models often exhibit limited domain transfer ability. Existing approaches to adaptation are unwieldy, such as requiring explicit supervision, complex model architectures, or massive external models. We present ABEL, a simple but effective unsupervised method to enhance passage retrieval in zero-shot settings. Our technique follows a straightforward loop: a dense retriever learns from supervision signals provided by a reranker, and subsequently, the reranker is updated based on feedback from the improved retriever. By iterating this loop, the two components mutually enhance one another's performance. Experimental results demonstrate that our unsupervised ABEL model outperforms both leading supervised and unsupervised retrievers on the BEIR benchmark. Meanwhile, it exhibits strong adaptation abilities to tasks and domains that were unseen during training. By either fine-tuning ABEL on labelled data or integrating it with existing supervised dense retrievers, we achieve state-of-the-art results.[1]

## 1 Introduction

Remarkable progress has been achieved in neural information retrieval through the adoption of the dual-encoder paradigm (Gillick et al., 2018), which enables efficient search over vast collections of passages by factorising the model such that the encoding of queries and passages are decoupled, and calculating the query-passage similarity using dot product. However, the efficacy of training dual-encoders heavily relies on the quality of labelled data, and these models struggle to maintain competitive performance on retrieval tasks where dedicated training data is scarce (Thakur et al., 2021).

Various approaches have been proposed to enhance dense retrievers (Karpukhin et al., 2020) in

zero-shot settings while maintaining the factorised dual-encoder structure, such as pre-training models on web-scale corpus (Izacard et al., 2022) and learning from cross-encoders through distillation (Qu et al., 2021). Other alternatives seek to trade efficiency for performance by using complex model architectures, such as fine-grained token interaction for more expressive representations (Santhanam et al., 2022) and scaling up the model size for better model capacity (Ni et al., 2022). Another line of work trains customised dense retrievers on target domains through query generation (Wang et al., 2022; Dai et al., 2023). This training paradigm is generally slow and expensive, as it employs large language models to synthesise a substantial number of high-quality queries.

In this paper, we present **ABEL**, an **A**lternating **B**ootstrapping training framework for unsupervised dense r**E**trieva**L**. Our method alternates the distillation process between a dense *retriever* and a *reranker* by switching their roles as *teachers* and *students* in iterations. On the one hand, the dense retriever allows for efficient retrieval due to its factorised encoding, accompanied by a compromised model performance. On the other hand, a reranker has no factorisation constraint, allowing for more fine-grained and accurate scoring, but at the cost of intractable searches. Our work aims to take advantage of both schools by equipping the dense retriever with accurate scoring by the reranker while maintaining search efficiency. Specifically, *i)* the more powerful but slower reranker is used to assist in the training of the less capable but more efficient retriever; *ii)* the dense retriever is employed to improve the performance of the reranker by providing refined training signals in later iterations. This alternating learning process is repeated to iteratively enhance both modules.

Compared with conventional bootstrapping approaches (Alberti et al., 2019; Zelikman et al., 2022), wherein the well-trained model itself is

---

[*]Now at Google DeepMind

[1]Source code is available at `https://github.com/Fantabulous-J/BootSwitch`.

used to discover additional solutions for subsequent training iterations, our method considers one model (i.e., teacher) as the training data generator to supervise another model (i.e., student), and their roles as teacher and student are switched in every next step. This mechanism naturally creates a mutual-learning paradigm to enable iterative bidirectional knowledge flow between the retriever and the reranker, in contrast to the typical single-step and unidirectional distillation where a student learns from a fixed teacher (Miech et al., 2021).

Through extensive experiments on various datasets, we observe that ABEL demonstrates outstanding performance in zero-shot settings by only using the basic BM25 model as an initiation. Additionally, both the retriever and reranker components involved in our approach can be progressively enhanced through the bootstrapping learning process, with the converged model outperforming more than ten prominent supervised retrievers and achieving state-of-the-art performance. Meanwhile, ABEL is efficient in its training process by exclusively employing sentences from raw texts as queries, rather than generating queries from large language models. The use of the simple dual-encoder architecture further contributes to its efficient operation. In summary, our contributions are:

1. We propose an iterative approach to bootstrap the ability of a dense retriever and a reranker without relying on manually-created data.

2. The empirical results on the BEIR benchmark show that the unsupervised ABEL outperforms a variety of prominent sparse and supervised dense retrievers. After fine-tuning ABEL using supervised data or integrating it with off-the-shelf supervised dense retrievers, our model achieves new state-of-the-art performance.

3. When applying ABEL on tasks that are unseen in training, we observe it demonstrates remarkable generalisation capabilities in comparison to other more sophisticated unsupervised dense retrieval methods.

4. To the best of our knowledge, we are the first to show the results that both dense retriever and cross-encoder reranker can be mutually improved in a closed-form learning loop, without the need for human-annotated labels.

## 2 Preliminary

Given a short text as a query, the passage-retrieval task aims at retrieving a set of passages that include the necessary information. From a collection of passages as the corpus, $\mathcal{P} = \{p_1, p_2, \cdots, p_n\}$, a *retriever* $\mathcal{D}$ fetches top-$k$ passages $\mathcal{P}_{\mathcal{D}}^k(q) = \{p_1, p_2, \cdots, p_k\}$ from $\mathcal{P}$ that are most relevant to a specific query $q$. Optionally, a reranker $\mathcal{R}$ is also employed to fine-grain the relevance scores for these retrieved passages.

### 2.1 Dense Retrieval Model

The dense retrieval model (retriever) encodes both queries and passages into dense vectors using a dual-encoder architecture (Karpukhin et al., 2020). Two distinct encoders are applied to transform queries and passages separately, then, a relevance score is calculated by a dot product,

$$\mathcal{D}(q, p; \theta) = \mathbf{E}(q; \theta_q)^\top \cdot \mathbf{E}(p; \theta_p), \qquad (1)$$

where $\mathbf{E}(\cdot; \theta)$ are encoders parameterised by $\theta_p$ for passages and $\theta_q$ for queries. The asymmetric dual-encoder works better than the shared-encoder architecture in our preliminary study. For efficiency, all passages in $\mathcal{P}$ are encoded offline, and an efficient nearest neighbour search (Johnson et al., 2021) is employed to fetch top-$k$ relevant passages.

### 2.2 Reranking Model

The reranking model (reranker) adopts a cross-encoder architecture, which computes the relevance score between a query and a passage by jointly encoding them with cross-attention. The joint encoding mechanism is prohibitively expensive to be deployed for large-scale retrieval applications. In practice, the joint-encoding model is usually applied as a reranker to refine the relevance scores for the results by the retriever. The relevance score by the reranker is formalised as,

$$\mathcal{R}(q, p; \phi) = \mathrm{FFN}(\mathbf{E}(q, p; \phi)), \qquad (2)$$

where $\mathbf{E}(\cdot, \cdot; \phi)$ is a pre-trained language model parameterised by $\phi$. In this work, we adopt the encoder of T5 (EncT5) (Liu et al., 2021) as $\mathbf{E}$. A start-of-sequence token is appended to each sequence, with its embedding fed to a randomly initialised single-layer feed-forward network (FFN) to calculate the relevance score.

## 3 Alternating Distillation

We propose an unsupervised alternating distillation approach that iteratively boosts the ability of a retriever and a reranker, as depicted in Fig. 1.

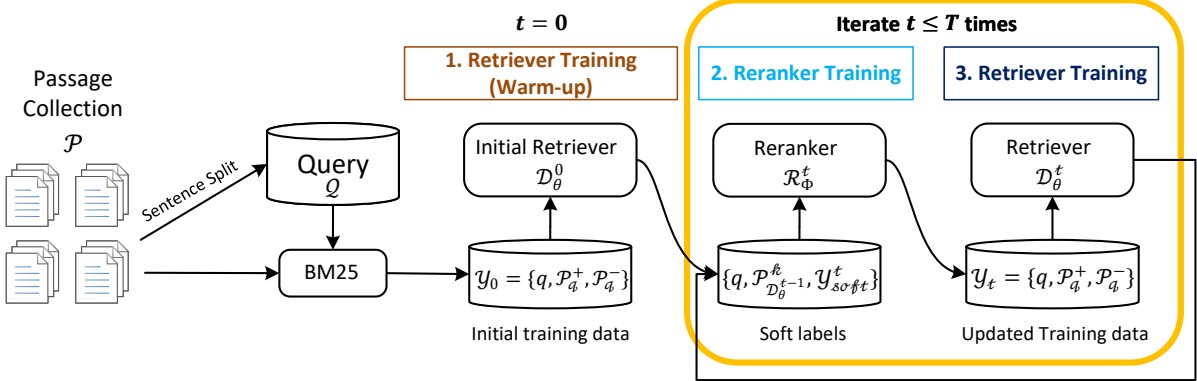

Figure 1: The overview of the alternating bootstrapping training approach for zero-shot dense retrieval.

Alg. 1 outlines the proposed method with three major steps. Our approach starts with warming up a retriever by imitating BM25. Subsequently, a recursive learning paradigm consisting of two steps is conducted: (1) training a reranker based on the labels extracted from the retriever by the last iteration; (2) refining the dense retriever using training signals derived from the reranker by the last step.

### 3.1 Retriever Warm-up

The training starts with constructing queries from raw texts (line 1) and training a warm-up dense retriever $\mathcal{D}_\theta^0$ by imitating an unsupervised BM25 model (lines 3-5).

**Query Construction**   We use a sentence splitter to chunk all passages into multiple sentences, then we consider these sentences as cropping-sentence queries (Chen et al., 2022). Compared to queries synthesised from fine-tuned query generators (Wang et al., 2022) or large language models (Dai et al., 2023), cropping-sentence queries *i)* can be cheaply scaled up without relying on supervised data or language models and *ii)* have been shown effectiveness in pre-training dense retrievers (Gao and Callan, 2022; Izacard et al., 2022).

**Training Data Initialisation**   The first challenge to our unsupervised method is how to extract effective supervision signals for each cropping-sentence query to initiate training. BM25 (Robertson and Zaragoza, 2009) is an unsupervised sparse retrieve model, which has demonstrated outstanding performance in low-resource and out-of-domain settings (Thakur et al., 2021). Specifically, for a given query $q \in \mathcal{Q}$, we use BM25 to retrieve the top-$k$ predictions $\mathcal{P}_{\text{BM25}}^k(q)$, among which the highest ranked $k^+ \in \mathbb{Z}^+$ passages are considered as

---

**Algorithm 1:** Alternating Bootstrapping Training for Zero-Shot Dense Retrieval

**Input** : Pre-trained language models (e.g., T5), and Passage collection $\mathcal{P}$.

1 Split passages into sentences as queries $\mathcal{Q}$.
2 /* **Step 1: Warm-up Retriever Training**   */
3 Retrieve top-$k$ predictions $\mathcal{P}_{\text{BM25}}^k(q)$ for each $q \in \mathcal{Q}$ using BM25 model.
4 Extract initial training data $\mathcal{Y}_0$ using Eq. 3 and 4.
5 Train a warm-up retriever $\mathcal{D}_\theta^0$ using $\mathcal{Y}_0$.
6 **for** $t \leftarrow 1$ *to* $T$ **do**
7     /* **Step 2: Reranker $\mathcal{R}_\phi$ Training**   */
8     Retrieve top-$k$ predictions $\mathcal{P}_{\mathcal{D}_\theta^{t-1}}^k(q)$ for each $q \in \mathcal{Q}$ using the most recent retriever $\mathcal{D}_\theta^{t-1}$.
9     Extract soft labels for each $(q, p)$ using $\mathcal{D}_\theta^{t-1}$: $\mathcal{Y}_{soft}^t = \{\mathcal{D}(q, p; \mathcal{D}_\theta^{t-1}) | p \in \mathcal{P}_{\mathcal{D}_\theta^{t-1}}^k(q)\}$.
10     Train an reranker $\mathcal{R}_\phi^t$ with $\mathcal{Y}_{soft}^t$ using Eq. 5.
11     /* **Step 3: Retriever $\mathcal{D}_\theta$ Training**   */
12     Rerank $\mathcal{P}_{\mathcal{D}_\theta^{t-1}}^k(q)$ using $\mathcal{R}_\phi^t$.
13     Extract updated training data $\mathcal{Y}_t$ from the reranking list using Eq. 3 and 4.
14     Fine-tune $\mathcal{D}_\theta^t$ using $\mathcal{Y}_t$ from $\mathcal{D}_\theta^0$.
15 **end for**
16 **return** $\mathcal{D}_\theta^T$.

---

positive $\mathcal{P}^+(q)$, while the bottom $k^- \in \mathbb{Z}^+$ passages are treated as hard negatives $\mathcal{P}^-(q)$, following Chen et al. (2022),

$$\mathcal{P}^+(q) = \{p | p \in \mathcal{P}_\mathcal{D}^k(q), r(p) \leq k^+\}, \qquad (3)$$

$$\mathcal{P}^-(q) = \{p | p \in \mathcal{P}_\mathcal{D}^k(q), r(p) \geq k - k^-\}, \quad (4)$$

where the initial $\mathcal{D}$ uses BM25 and $r(p)$ means the rank of a passage $p$. Then, we train a warm-up retriever based on these extracted training examples.

### 3.2 Iterative Bootstrapping Training

We iteratively improve the capability of the retriever and the reranker by alternating their roles as teacher and student in each iteration.

In the $t$-th iteration, we first use the most recent retriever $\mathcal{D}_\theta^{t-1}$ to retrieve top-$k$ passages $\mathcal{P}_{\mathcal{D}_\theta^{t-1}}^k(q)$ that are most relevant to each query $q \in \mathcal{Q}$ and generate soft labels $\mathcal{D}(q, p; \theta^{t-1})$ for each $(q, p)$ pair accordingly. Then we use all such soft labels to train a reranker $\mathcal{R}_\phi^t$ as described in Alg. 1 lines 8-10 and §3.3.1. The second step is to train the $t$-th retriever. We employ $\mathcal{R}_\phi^t$ to rerank $\mathcal{P}_{\mathcal{D}_\theta^{t-1}}^k(q)$ to obtain a refined ranking list, from which updated supervision signals are derived as Alg. 1 lines 12-13. We train a new retriever $\mathcal{D}_\theta^t$ with these examples as discussed in Alg. 1 line 14 and §3.3.2. Training iterations are repeated until no performance improvement is observed. Note that, in order to mitigate the risk of overfitting the retriever, for all iterations we refine the warm-up retriever $\mathcal{D}_\theta^0$ using the newest training examples but refresh top-$k$ predictions and update soft labels for reranker training with the improved retriever. Similarly, to avoid the accumulation of errors in label generation, we re-initialise the reranker using pre-trained language models at the start of each iteration, rather than fine-tuning the model obtained from the last iteration. Please refer to Table 3 for the empirical ablations.

### 3.3 Retriever and Reranker Training

In each iteration, we fine-tune the reranker and then the retriever as follows.

#### 3.3.1 Reranker Training

The reranker is trained to discriminate positive samples from hard negative ones using a cross-entropy (CE) loss,

$$\mathcal{L}_{\text{CE}} = -\log \frac{\exp(s(q, p^+; \phi))}{\sum_{p \in \{p^+\} \cup \mathcal{P}_q^-} \exp(s(q, p; \phi))}.$$

In our preliminary experiments, we observed that using hard labels to train the rereanker yields poor results. The reason is that hard labels use binary targets by taking one query-passage pair as positive and the rest pairs as negative, failing to provide informative signals for discerning the subtle distinctions among passages. To address this problem, we consider using the soft labels generated by a dense retriever $\mathcal{D}_\theta$ to guide the reranker training. These labels effectively capture the nuanced semantic relatedness among multiple relevant passages. Specifically, we employ the KL divergence loss:

$$\mathcal{L}_{\text{KL}} = \mathbb{D}_{\text{KL}}(S(\cdot|q, \mathcal{P}_q, \phi) || T(\cdot|q, \mathcal{P}_q, \theta)),$$

where $\mathcal{P}_q$ is the retrieved set of passages regarding to query $q$, which are sampled from $\mathcal{P}_{\mathcal{D}_\theta^{t-1}}^k(q)$.

$T(\cdot|q, \mathcal{P}_q, \theta)$ and $S(\cdot|q, \mathcal{P}_q, \phi)$ are the distributions from the teacher retriever and the student reranker, respectively. Our preliminary experiment shows that adding noise to the reranker's inputs (e.g., word deletion) can further enhance the performance. For more details, please refer to Table 4 in Appendix B.

#### 3.3.2 Retriever Training

For each query $q$, we randomly sample one positive $p^+$ and one hard negative $p^-$ from $\mathcal{P}^+(q)$ (Eq. 3) and $\mathcal{P}^-(q)$ (Eq. 4), respectively. In practice, we use in-batch negatives (Karpukhin et al., 2020) for efficient training. The dense retriever $\mathcal{D}_\theta$ is trained by minimising the negative log-likelihood of the positive passages using contrastive learning (CL) (Hadsell et al., 2006),

$$\mathcal{L}_{\text{CL}} = -\log \frac{\exp(s(q_i, p_i^+; \theta))}{\sum_{j=1}^{|\mathcal{B}|} \sum_{p \in \{p_j^+, p_j^-\}} \exp(s(q_i, p; \theta))}.$$

where $|\mathcal{B}|$ is the size of a batch $\mathcal{B}$. Similarly, we find that injecting noise into the inputs of the retriever during training results in improved performance, as demonstrated in Table 4 in Appendix B. We believe that manipulating words in queries and passages can be seen as a smoothing method to facilitate the learning of generalisable matching signals (He et al., 2020), therefore, preventing the retriever from simply replying on examining the text overlaps between the cropping-sentence query and passages. Additionally, adding noise tweaks the semantics of texts, which encourages the dense retriever to acquire matching patterns based on lexical overlaps in addition to semantic similarities.

### 3.4 Discussion

The novelty of our approach lies in two aspects: *i)* We introduce a separate reranker model for training label refinement. This allows the retriever training to benefit from an advanced reranker, which features a more sophisticated cross-encoding architecture. This design is effective compared to training the retriever on labels extracted from its own predictions (see the blue line in Figure 5(a)); *ii)* We create a mutual-learning paradigm through iterative alternating distillation. In our case, we consider the retriever as a proposal distribution generator, which selects relatively good candidates, and the reranker as a reward scoring model, which measures the quality of an example as a good answer. At each iteration, the improved reranker can be used to correct the predictions from the retriever,

| Setting | Supervised | | | | | | | | | | | Unsupervised | | | | |
|---|---|---|---|---|---|---|---|---|---|---|---|---|---|---|---|---|
| Model | ANCE | Contriever | RetroMAE | GTR-XXL | GPL† | PTR† | COCO-DR | ColBERTv2 | SPLADE++ | DRAGON+ | **ABEL-FT** | BM25 | REALM | SimCSE | Contriever | **ABEL** |
| Model Size | 110M | 110M | 110M | 4.8B | 66M | 110M | 110M | 110M | 110M | 220M | 220M | - | 110M | 355M | 110M | 220M |
| QGen. Size | - | - | - | - | 220M | 137B | - | - | - | 220M | - | - | - | - | - | - |
| Retriever Type | dense | dense | dense | dense | dense | dense | dense | mul-vec | sparse | dense | dense | sparse | dense | dense | dense | dense |
| Distillation | ✗ | ✗ | ✗ | ✗ | ✓ | ✗ | ✗ | ✓ | ✓ | ✓ | ✓ | ✗ | ✗ | ✗ | ✗ | ✓ |
| TREC-COVID | 64.9 | 59.6 | 77.2 | 50.1 | 70.0 | 72.7 | **78.9** | 73.8 | 71.1 | 75.9 | 76.5 | 65.6 | 10.0 | 38.6 | 26.2 | **72.7** |
| BioASQ | 30.9 | 38.3 | 42.1 | 32.4 | 44.2 | - | 42.9 | - | **50.4** | 43.3 | 45.4 | **46.5** | 23.0 | 6.1 | 30.6 | 41.3 |
| NFCorpus | 23.5 | 32.8 | 30.8 | 34.2 | 34.5 | 33.4 | **35.5** | 33.8 | 34.5 | 33.9 | 35.1 | 32.5 | 20.9 | 14.0 | 32.3 | **33.8** |
| NQ | 44.4 | 49.5 | 51.8 | 56.8 | 48.3 | - | 50.5 | 56.2 | 54.4 | 53.7 | 50.2 | 32.9 | 21.8 | 12.6 | 26.9 | **42.0** |
| HotpotQA | 45.1 | 63.8 | 63.5 | 59.9 | 58.2 | 60.4 | 61.6 | 66.7 | **68.6** | 66.2 | 65.7 | 60.3 | 40.5 | 23.3 | 45.5 | **61.9** |
| FiQA-2018 | 29.5 | 32.9 | 31.6 | **46.7** | 34.4 | 40.4 | 31.7 | 35.6 | 35.1 | 35.6 | 34.3 | 23.6 | 6.1 | 14.8 | 25.0 | **31.1** |
| Signal-1M (RT) | 24.9 | 19.9 | 26.5 | 27.3 | 27.6 | - | 27.1 | - | 29.6 | **30.1** | 28.0 | **33.0** | 13.8 | 21.4 | 23.7 | 30.8 |
| TREC-NEWS | 38.4 | 42.8 | 42.8 | 34.6 | 42.1 | - | 40.3 | - | 39.4 | 44.4 | **45.4** | 39.8 | 26.1 | 25.7 | 39.4 | **48.0** |
| Robust04 | 39.2 | 47.6 | 44.7 | 50.6 | 43.7 | - | 44.3 | - | 45.8 | 47.9 | 50.0 | 40.8 | 20.7 | 30.0 | 34.5 | **48.4** |
| ArguAna | 41.9 | 44.6 | 43.3 | 54.0 | 55.7 | 53.8 | 49.3 | 46.3 | 52.1 | 46.9 | **56.9** | 31.5 | 18.7 | 45.6 | 40.6 | **50.5** |
| Touché-2020 | 24.0 | 23.0 | 23.7 | 25.6 | 25.5 | **26.6** | 23.8 | 26.3 | 24.4 | 26.3 | 19.5 | **36.7** | 4.9 | 11.6 | 21.7 | 29.5 |
| CQADupStack | 29.8 | 34.5 | 34.7 | **39.9** | 35.7 | - | 37.0 | - | 34.1 | 35.4 | 36.9 | 29.9 | 10.5 | 20.2 | 28.2 | **35.0** |
| Quora | 85.2 | 86.5 | 84.7 | **89.2** | 83.6 | - | 86.7 | 85.2 | 81.4 | 87.5 | 84.5 | 78.9 | 39.8 | 81.5 | 83.2 | **83.9** |
| DBPedia | 28.1 | 41.3 | 39.0 | 40.8 | 38.4 | 36.4 | 39.1 | **44.6** | 44.2 | 41.7 | 41.4 | 31.3 | 21.4 | 13.7 | 29.9 | **37.5** |
| SCIDOCS | 12.2 | 16.5 | 15.0 | 16.1 | 16.9 | 16.3 | 16.0 | 15.4 | 15.9 | 15.9 | **17.4** | 15.8 | 7.0 | 7.4 | 15.3 | **17.5** |
| FEVER | 66.7 | 75.8 | 77.4 | 74.0 | 75.9 | 76.2 | 75.1 | 78.5 | **79.6** | 78.1 | 74.1 | 75.3 | 42.7 | 20.1 | 61.9 | 74.1 |
| Climate-FEVER | 20.0 | 23.7 | 23.2 | **26.7** | 23.5 | 21.4 | 21.1 | 17.6 | 22.7 | 22.7 | 21.8 | 21.3 | 15.4 | 17.6 | 17.2 | **25.3** |
| SciFact | 51.0 | 67.7 | 65.3 | 66.2 | 67.4 | 62.3 | 70.9 | 69.3 | 69.9 | 67.9 | **72.6** | 66.5 | 46.1 | 38.5 | 64.6 | **73.5** |
| Avg. PTR-11 | 37.0 | 43.8 | 44.5 | 44.9 | 45.5 | 45.5 | 45.7 | 46.2 | **47.1** | 46.5 | 46.9 | 36.1 | 21.0 | 27.9 | 30.3 | **46.1** |
| Avg. BEIR-13 | 41.3 | 47.5 | 48.2 | 49.3 | 48.6 | - | 49.2 | 49.9 | **50.3** | 50.2 | 50.0 | 44.0 | 22.7 | 26.1 | 37.7 | **48.7** |
| Avg. All | 38.9 | 44.5 | 45.4 | 45.8 | 45.9 | - | 46.2 | - | 47.4 | 47.4 | **47.5** | 42.3 | 21.6 | 24.6 | 35.9 | **46.5** |

Table 1: Zero-shot retrieval results on BEIR (nDCG@10). The best and second-best results are marked in **bold** and underlined. Methods that train dedicated models for each of the datasets are noted with †. Highlighted rows represent semantic relatedness tasks. QGen. means query generator. Please refer to Appendix A.2 for baseline details.

therefore reducing the number of inaccurate labels in retriever training data. Meanwhile, the improved retriever is more likely to include more relevant passages within its top-$k$ predictions and provide more nuanced soft labels, which can enhance the reranker training and thus, increase the chance of finding correct labels by the reranker in the next iteration. As the training continues, (1) the candidates of correct answers can be narrowed down by the retriever (i.e., better proposal distributions), as shown in Fig. 4 and (2) the accuracy of finding correct answers from such sets can be increased by the reranker, as shown in Fig. 2(b). Overall, our framework (i.e., ABEL) facilitates synergistic advancements in both components, ultimately enhancing the overall effectiveness of the retrieval system.

## 4 Experiments

### 4.1 Dataset

Our method is evaluated on the BEIR benchmark (Thakur et al., 2021), which contains 18 datasets in multiple domains such as wikipedia and biomedical, and with diverse task formats including question answering, fact verification and paraphrase retrieval. We use nDCG@10 (Järvelin and Kekäläinen, 2002) as our primary metric and the average nDCG@10 score over all 18 datasets is used for comprehensive comparison between models. BEIR-13 (Formal et al., 2021) and PTR-11 (Dai et al., 2023) are two subsets of tasks, where BEIR-13 excludes CQADupStack, Robust04, Signal-1M, TREC-NEWS, and BioASQ from the calculation of average score and PTR-11 further removes NQ and

Quora. We also partition the datasets into *query search* (e.g., natural questions) and *semantic relatedness* (e.g., verification) tasks, in accordance with Santhanam et al. (2022),

## 4.2 Experimental Settings

We use `Contriever` to initialise the dense retriever. The passages from all tasks in BEIR are chunked into cropping-sentence queries, on which a *single* retriever ABEL is trained. We initialise the reranker from `t5-base-lm-adapt` (Raffel et al., 2020) and train a *single* reranker ABEL-Rerank similarly. We conduct the alternating distillation loop for three iterations and observe that the performance of both the retriever and reranker converges, and we take ABEL in iteration 2 and ABEL-Rerank in iteration 3 for evaluation and model comparison. We further fine-tune ABEL on MS-MARCO (Bajaj et al., 2016) to obtain ABEL-FT, following the same training recipe outlined in Izacard et al. (2022). In addition, we also evaluate ABEL and ABEL-Rerank on various datasets that are unseen during training to test the generalisation ability. More details are in Appendix A.

## 4.3 Experimental Results

**Retriever Results** Both unsupervised and supervised versions of our model are compared with a range of corresponding baseline models in Table 1. The unsupervised ABEL outperforms various leading dense models, including models that are supervised by MS-MARCO training queries. The supervised model, ABEL-FT, achieves 1.0% better overall performance than ABEL. ABEL-FT also surpasses models using the target corpora for pre-training (COCO-DR), employing fine-grained token interaction (ColBERTv2), using sparse representations (SPLADE++), with larger sizes (GTR-XXL), and using sophisticated training recipes with diverse supervision signals (DRAGON+).

Considering semantic relatedness tasks, such as Signal-1M, Climate-FEVER, and SciFact, ABEL generally achieves results superior to other supervised dense retrievers. For query-search tasks, particularly question-answering tasks like NQ and HotpotQA, ABEL underperforms many dense retrievers. We attribute such outcomes to the differences in query styles. Semantic relatedness tasks typically use short sentences as queries, which aligns with the cropping-sentence query format employed in our work. However, query-search tasks often involve natural questions that deviate significantly

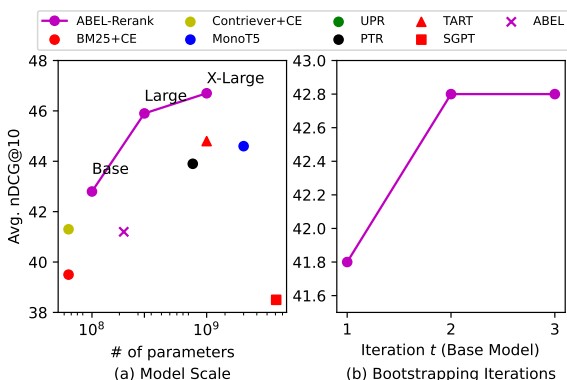

Figure 2: Reranking results on BEIR, with (a) the comparison of models with various sizes and (b) the accuracy of the reranker in iteration $t$ using the base model.

from the cropping-sentence queries, and such format mismatch leads to the inferior performance of ABEL. For lexical matching tasks, such as Touché-2020, ABEL surpasses the majority of dense retrievers by a considerable margin. We attribute this success to the model's ability to capture salient phrases, which is facilitated by learning supervision signals from BM25 in retriever warm-up training and well preserved in the following training iterations. Finally, ABEL outperforms GPL and PTR, even though they have incorporated high-quality synthetic queries and cross-encoder distillation in training. This observation demonstrates that a retriever can attain promising results in zero-shot settings without relying on synthetic queries.

For the supervised setting, the performance of ABEL can be improved by fine-tuning it on supervised data in dealing with natural questions. The major performance gains are from query-search tasks and semantic relatedness tasks, which involve human-like queries, such as NQ (42.0 to 50.2) and DBPedia (37.5 to 41.4). On other datasets with short sentences as queries (e.g., claim verification), the performance of ABEL-FT degrades but is comparable to other supervised retrievers. This limitation can be alleviated by combining ABEL and ABEL-FT, thereby achieving performance improvements on both types of tasks, as illustrated in the last two bars of Figure 3.

**Reranker Results** Figure 2 shows the averaged reranking performance on 9 subsets of BEIR, excluding FEVER and HotpotQA from PTR-11. As shown in Figure 2(b), the reranker is able to achieve improvements as the bootstrapping iteration progresses, providing strong evidence for the effec-

| | AMB | WQA | GAT | LSO | CSP | Avg. |
|---|---|---|---|---|---|---|
| BM25 | 96.3 | 57.7 | 64.1 | **31.8** | 27.7 | 55.5 |
| REALM | 88.4 | 58.2 | 43.0 | 7.2 | 5.9 | 40.5 |
| SimCSE | 92.3 | 48.9 | 41.7 | 9.3 | 13.8 | 41.2 |
| Contriever | 96.0 | 45.2 | 67.2 | 18.3 | 28.4 | 51.0 |
| **ABEL** | **97.4** | **69.6** | **69.3** | 22.3 | **35.4** | **58.8** |

Table 2: Zero-shot cross-task retrieval results of unsupervised models based on nDCG@10. The best and second-best results are marked in **bold** and underlined. The tasks are AMB=AmbigQA, WQA=WikiQA, GAT=GooAQ-Technical, LSO=LinkSO-Python, CSP=CodeSearchNet-Python.

tiveness of our iterative alternating distillation approach. The final reranker model, ABEL-Rerank (i.e., $t = 3$), as illustrated in Figure 2(a), enhances the performance of ABEL by 1.6%, surpassing supervised models of similar parameter sizes. It is noteworthy that ABEL-Rerank outperforms unsupervised SGPT (Muennighoff, 2022) and zero-shot UPR (Sachan et al., 2022), despite having much fewer parameters, and is comparable to PTR, which creates dedicated models for each task and employs high-quality synthetic queries. With the increase in model size, we consistently observe improvements with ABEL-Rerank and it outperforms TART (Asai et al., 2022) and MonoT5 (Nogueira et al., 2020) using only a fraction of the parameters. This finding demonstrates the potential of incorporating more capable rerankers to train better retrievers. We leave this exploration to future work. Please refer to Tables 5 and 7 in Appendix D for further details.

**Cross-Task Results** As shown in Table 2, when directly evaluating ABEL on various tasks unseen during training, it consistently achieves significant improvements over Contriever (+7.8%) and outperforms BM25 and other advanced unsupervised retrievers. These results show that ABEL is capable of capturing matching patterns between queries and passages that can be effectively generalised to unseen tasks, instead of memorising training corpus to achieve good performance. Please refer to Appendix E for more results.

### 4.4 Analysis

**Pre-trained Models** Table 3 #1 compares our approach with DRAGON when using different pre-trained models for initialisation. ABEL outperforms DRAGON consistently using aligned pre-trained models, with up to +0.6% gains. Moreover, our method exhibits continued improvement as we

| #1. *Different Pre-trained Models* | | |
|---|---|---|
| Initialisation | DRAGON | ABEL |
| BERT | 46.8 | **47.2** |
| Contriever | - | 47.5 |
| RetroMAE | 47.4 | **48.0** |

| #2. *Different Training Corpus* | | |
|---|---|---|
| Corpus | Avg. | $\Delta$ |
| Wikipedia & CCNet | 35.9 | - |
| MS-MARCO | 41.0 | +5.1 |
| BEIR | **45.1** | +9.2 |

| #3. *Model Re-initialisation* | | |
|---|---|---|
| Strategy | w/o re-init | w/ re-init |
| Avg. | 45.7 | **46.5** |

Table 3: Ablations on pre-trained models, the corpus used for training, and the effects of model re-initialisation. The average performance (nDCG@10) on BEIR is reported. Note that for training corpus ablation, results for retrievers trained with iteration $t = 1$ are reported. Wikipedia & CCNet are the training corpora of Contriever and $\Delta$ indicates performance gains over Contriever.

use more advanced pre-trained checkpoints. This demonstrates that our approach is orthogonal to existing unsupervised pre-training methods, and further gains are expected when more sophisticated pre-trained models are available.

**Training Corpus** We compare models trained using cropped sentences from different corpus as the training data. As shown in Table 3 #2, the method trained using MS-MARCO corpus is significantly better than the vanilla Contriever (+5.1%) but is inferior to the one using diverse corpus from BEIR, and we believe that the diverse corpora we used in training is one of the factors to our success.

**Model Re-initialisation** We use re-initialisation to avoid the accumulation from biased errors in early iterations. At the start of each iteration, we re-initialise the retriever with the warm-up retriever and the reranker using pre-trained language models, respectively, rather than continuously fine-tuning the models obtained from the last iteration. Table 3 #3 shows the overall performance is increased by 0.8% using this re-initialisation technology.

**Combination with Supervised Models** We investigate whether ABEL can advance supervised retrievers. We merge the embeddings from ABEL with different supervised models through concatenation. Specifically, for a given query $q$ and passage $p$, and

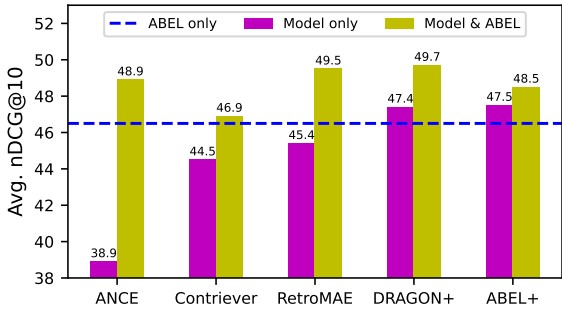

Figure 3: The comparison of combining ABEL and supervised dense retrievers (Model & ABEL) on BEIR.

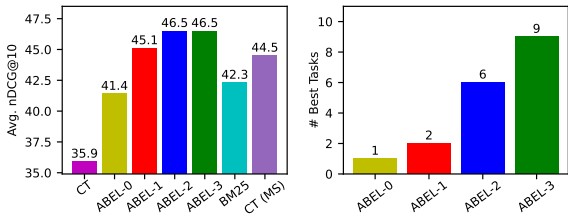

Figure 4: The effect of bootstrapping the retrieval ability on BEIR. CT and CT (MS) indicate Contriever and Contriever (MS), respectively. # Best Tasks: the number of tasks on which the models perform the best.

dense retrievers $\mathbf{E}_q^i$ and $\mathbf{E}_p^i$, we compute the relevance score as $s(q, p) = [\mathbf{E}_q^1, \mathbf{E}_p^1]^\top \cdot [\mathbf{E}_q^2, \mathbf{E}_p^2] = \sum_{i=1}^{2} \mathbf{E}_q^{i\top} \cdot \mathbf{E}_p^i$. The results shown in Figure 3 indicate ABEL can be easily integrated with other models to achieve significant performance improvement, with an average increase of up to 4.5% on RetroMAE and even a 1% gain on ABEL-FT. Besides, the benefit is also remarkable (+10%) when combining ABEL with weak retrievers (i.e., ANCE). Overall, we observe that the ensembling can result in performance gains in both query-search and semantic-relatedness tasks, as demonstrated in Table 11 in Appendix F.

**Effect of Bootstrapping** Figure 4 presents the comparison of ABEL's performance throughout all bootstrapping iterations against the BM25 and the supervised Contriever. We observe that the accuracy of the retriever consistently improves as the training iteration $t$ progresses. Specifically, ABEL matches the performance of the supervised Contriever on the first iteration, and further gains are achieved with more iterations $t \leq 2$. The performance converges at iteration 3, where the results on six tasks are inferior to those achieved at iteration 2. Please refer to Figure 6 in Appendix G for results on each individual task.

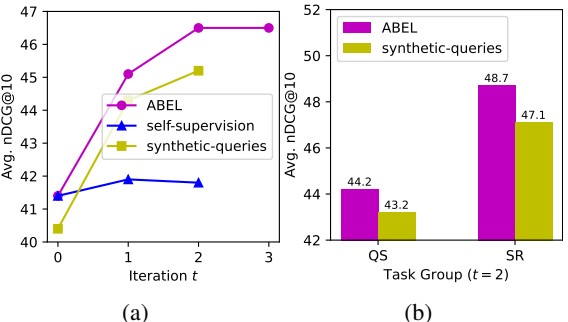

Figure 5: Results on BEIR by removing the reranker component or using synthetic queries for training, where QS=Query-Search and SR=Semantic-Relatedness.

**Self Supervision** We investigate the necessity of taking the reranker as an expert in ABEL. Specifically, we use the top-$k$ predictions of the latest retriever to extract training data at each iteration, instead of employing a separate reranker (i.e., without lines 8-10 in Alg.1). The blue line in Figure 5(a) indicates that the retriever struggles to improve when using itself as the supervisor. By investigating a small set of generated labels, we notice the extracted positive passages for most queries quickly converge to a stable set, failing to offer new signals in the new training round. This highlights the essential role of the expert reranker, which iteratively provides more advanced supervision signals.

**Synthetic Queries** We assess the utility of synthetic queries in our approach by replacing cropping-sentence queries with synthetic queries from a query-generator fine-tuned on MS-MARCO.[2] The results in Figure 5(a) show that using synthetic queries is less effective and exhibits similar trends, with the performance improving consistently as the iterative alternating distillation progresses.[3] Splitting task groups, we observe synthetic queries yield a larger performance drop on semantic-relatedness tasks than query-search tasks, in Figure 5(b). We attribute this disparity to the stylistic differences between the training and test queries. Synthetic queries exhibit similarities to natural questions in terms of style, akin to those found in question-answering datasets. In contrast, semantic relatedness tasks usually involve short-sentence queries (e.g., claim) that are closer to cropping-sentence queries. This finding empha-

---

[2]https://public.ukp.informatik.tu-darmstadt.de/kwang/gpl/generated-data/beir
[3]Using large language models for query generation may yield better results. We leave this exploration to future work.

sises the importance of aligning the formats of training queries with test queries in zero-shot settings. Please refer to Figure 7 in Appendix H for results comparison in each individual task.

## 5 Related Work

**Neural Information Retrieval**   Neural retrievers adopt pre-trained language models and follow a dual-encoder architecture (Karpukhin et al., 2020) to generate the semantic representations of queries and passages and then calculate their semantic similarities.   Some effective techniques have been proposed to advance the neural retrieval models, such as hard negative mining (Xiong et al., 2021), retrieval-oriented pre-training objectives (Izacard et al., 2022), and multi-vector representations (Khattab and Zaharia, 2020). All of these approaches require supervised training data and suffer from performance degradation on out-of-domain datasets.  Our work demonstrates the possibility that an unsupervised dense retriever can outperform a diverse range of state-of-the-art supervised methods in zero-shot settings.

**Zero-shot Dense Retrieval**   Recent research has demonstrated that the performance of dense retrievers under out-of-domain settings can be improved through using synthetic queries (Ma et al., 2021; Gangi Reddy et al., 2022; Dai et al., 2023). Integrating distillation from cross-encoder rerankers further advances the current state-of-the-art models (Wang et al., 2022).  However, all of these methods rely on synthetic queries, which generally implies expensive inference costs on the usage of large language models. Nonetheless, the quality of the synthetic queries is worrying, although it can be improved by further efforts, such as fine-tuning the language model on high-quality supervised data (Wei et al., 2022). In contrast, our work does not have such reliance, and thus offers higher training efficiency. We show that effective large-scale training examples can be derived from raw texts in the form of cropping-sentence queries (Chen et al., 2022), and their labels can be iteratively refined by the retriever and the reranker to enhance the training of the other model.

**Iterated Learning**   Iterated Learning refers to a family of algorithms that iteratively use previously learned models to update training labels for subsequent rounds of model training. These algorithms may also involve filtering out examples of low qual-

ity by assessing whether the solution aligns with the desired objective (Alberti et al., 2019; Zelikman et al., 2022; Dai et al., 2023).  This concept can also be extended to include multiple models. One method is the iterated expert introduced by Anthony et al. (2017), wherein an *apprentice* model learns to imitate an *expert* and the *expert* builds on improved *apprentice* to find better solutions. Our work adapts such a paradigm to retrieval tasks in a zero-shot manner, where a notable distinction from previous work is the novel iterative *alternating distillation* process. For each iteration, the roles of the retriever and the reranker as *apprentice* and *expert* are alternated, enabling bidirectional knowledge transfer to encourage mutual learning.

## 6 Conclusion

In this paper, we introduce ABEL, an unsupervised training framework that iteratively improves both retrievers and rerankers. Our method enhances a dense retriever and a cross-encoder reranker in a closed learning loop, by alternating their roles as teachers and students. The empirical results on various tasks demonstrate that this simple technique can significantly improve the capability of dense retrievers without relying on any human-annotated data, surpassing a wide range of competitive sparse and supervised dense retrievers. We believe that ABEL is a generic framework that could be easily combined with other retrieval augmenting techniques, and benefits a range of downstream tasks.

## Limitations

The approach proposed in this work incurs additional training costs on the refinement of training labels and the iterative distillation process when compared to standard supervised dense retriever training. The entire training pipeline requires approximately one week to complete on a server with 8 A100 GPUs. This configuration is relatively modest according to typical academic settings.

   We focus on the standard dual-encoder paradigm, and have not explored other more advanced architectures, such as ColBERT, which offer more expressive representations. We are interested in investigating whether incorporating these techniques would yield additional benefits to our approach. Furthermore, existing research (Ni et al., 2022) has demonstrated that increasing the model size can enhance the performance and generalisability of dense retrievers. Our analysis also shows that

scaling up the model size improves reranking performance. Therefore, we would like to see whether applying the scaling effect to the retriever side can result in further improvement of the performance.

Moreover, we mainly examined our method on the BEIR benchmark. Although BEIR covers various tasks and domains, there is still a gap to industrial scenarios regarding the diversity of the retrieval corpora, such as those involving web-scale documents. We plan to explore scaling up the corpus size in our future work. Additionally, BEIR is a monolingual corpus in English, and we are interested in validating the feasibility of our method in multi-lingual settings.

## Acknowledgements

We thank the anonymous reviewers for their helpful feedback and suggestions. The first author is supported by the Graduate Research Scholarships funded by the University of Melbourne. This work was funded by the Australian Research Council, Discovery grant DP230102775. This research was undertaken using the LIEF HPCGPGPU Facility hosted at the University of Melbourne, which was established with the assistance of LIEF Grant LE170100200.

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

## A  Experimental Settings

### A.1  Implementation Details

For training queries, we divide the passages in the corpus of each task in BEIR into individual sentences, treating them as cropping-sentence queries. In datasets with a large corpus size (e.g., Climate-FEVER), we randomly select 2 million sentences for training. However, for datasets with fewer passages (~5k-50k), we use all of them for training. For each query, we follow Chen et al. (2022) to extract its positive and negative passages from the top-$k$ predictions of a specific retriever, where the top-10 are considered as positives while passages ranked between 46 and 50 are regarded as negatives. For the reranking process in line 12 of Algorithm 1, we always rerank the top 100 retrievals returned by a specific retriever, from which refined labels are extracted using the above rules.

We use `Contriever`[4] to initialise the dense retriever and train it for 3 epochs on 8 A100 GPUs, with a per-GPU batch size 128 and learning rate $3 \times 10^{-5}$. Each query is paired with one positive and one negative passage together with in-batch negatives for efficient training. A *single* retriever `ABEL` is trained on the union of queries from all tasks on BEIR. We initialise the reranker from `t5-base-lm-adapt` (Raffel et al., 2020) checkpoint.[5] For each query, we sample one positive and 7 negative passages. Similarly, we train a *single* reranker `ABEL-Rerank` using a batch size 64 and learning rate $3 \times 10^{-5}$ for 20k steps, with roughly 1.2k steps on each task. We conduct the iterative alternating distillation for three iterations and take `ABEL` in iteration 2 and `ABEL-Rerank` in iteration 3 for evaluation and result comparison. We set the maximum query and passage lengths to 128 and 256 for training and set both input lengths to 512 for evaluation.

We further fine-tune `ABEL` on MS-MARCO (Bajaj et al., 2016) using in-batch negatives for 10k steps, with a batch size 1024 and learning rate $1 \times 10^{-5}$. The maximum query and passage lengths for training are set to 32 and 128, respectively. Following Izacard et al. (2022), we first train an initial model with each query paired with one gold positive and a randomly-sampled negative. We then mine hard negatives with this model and retrain a second model `ABEL-FT` in the same manner but

| Method | Avg. nDCG@10 |
|---|---|
| Initial Retriever ($\mathcal{D}_\theta^0$) | 40.2 |
| + noise | **41.4** |
| Initial Reranker ($t = 1$) | 41.9 |
| + noise | 43.7 |
| + soft labels | 45.6 |
| + noise & soft labels | **47.1** |

Table 4: Study on the effects of injecting input noise and using soft labels on model training. Average performance on the BEIR benchmark is reported.

with a hard negative 10% of the time.

### A.2  Baseline Retrievers

We compare our method with a wide range of unsupervised and supervised models. Unsupervised models include: (1) BM25 (Robertson and Zaragoza, 2009); (2) Contriever (Izacard et al., 2022) that is pre-trained on unlabelled text with contrastive learning; (3) SimCSE[6] (Gao et al., 2021) that uses contrastive learning to learn unsupervised sentence representations by taking the encodings of a single sentence with different dropout masks as positive pairs; (4) REALM[7] (Guu et al., 2020) that trains unsupervised dense retrievers using the masked language modelling signals from a separate reader component. Supervised models include (1) ANCE (Xiong et al., 2021) trained on MS-MARCO with self-mined dynamic hard negatives; (2) Contriever (MS) and COCO-DR (Yu et al., 2022) that are first pretrained on unlabelled corpus with contrastive learning and then fine-tuned on MS-MARCO; (3) RetroMAE (Xiao et al., 2022) uses masked auto-encoding for model pre-training and MS-MARCO for fine-tuning; (4) GTR-XXL (Ni et al., 2022), ColBERTv2 (Santhanam et al., 2022) and SPLADE++ (Formal et al., 2022) that use significantly larger model size, multi-vector and sparse representations, along with distillation from cross-encoder on MS-MARCO; (5) DRAGON+ (Lin et al., 2023) that learns progressively from diverse supervisions provided by above models on MS-MARCO with both crop-sentence and synthetic queries; (6) QGen: GPL (Wang et al., 2022) and PTR (Dai et al., 2023) that create customised models for each target task by using synthetic queries and pseudo relevance labels.

---

[4] https://huggingface.co/facebook/contriever
[5] https://huggingface.co/google/t5-base-lm-adapt

[6] https://huggingface.co/princeton-nlp/unsup-simcse-roberta-large
[7] https://huggingface.co/google/realm-cc-news-pretrained-embedder

| Method | top-$\mathbb{K}$ | Avg. Performance | | | |
|---|---|---|---|---|---|
| | | TART-9 | PTR-11 | BEIR-13 | All |
| *Supervised by MS MARCO* | | | | | |
| BM25+CE | 100 | 39.5 | 46.2 | 49.5 | 47.6 |
| Contriever+CE | 100 | 41.3 | 46.6 | 50.2 | - |
| MonoT5 (3B) | 1000 | 44.6 | 51.1 | - | - |
| *Zero-shot Domain Adaption* | | | | | |
| UPR (3B) | 1000 | 37.8 | 42.7 | 46.2 | - |
| PTR$^\dagger$ (110M×11) | 200 | 43.9 | 49.9 | - | - |
| TART (1.24B) | 100 | 44.8 | - | - | - |
| *Unsupervised* | | | | | |
| SGPT (6.1B) | 100 | 38.5 | 44.3 | 46.7 | 46.2 |
| **ABEL-Rerank (110M)** | 100 | 42.8 | 48.5 | 50.7 | 48.3 |
| **ABEL-Rerank (335M)** | 100 | 45.9 | 51.3 | 53.7 | 51.3 |
| **ABEL-Rerank (1.24B)** | 100 | **46.7** | **52.3** | **54.9** | **52.4** |

Table 5: The comparison of state-of-the-art rerankers with ABEL-Rerank on BEIR benchmark using nDCG@10. The averaged results of different experiments are reported. Methods that train dedicated models for corresponding datasets are noted with $^\dagger$. TART-9 excludes FEVER and HotpotQA from PTR-11.

| Model | FQ | SF | TC | CQ | RB | Avg. |
|---|---|---|---|---|---|---|
| SimCSE | 26.7 | 55.0 | 68.3 | 29.0 | 37.9 | 43.4 |
| ICT | 27.0 | 58.5 | 69.7 | 31.3 | 37.4 | 44.8 |
| MLM | 30.2 | 60.0 | 69.5 | 30.4 | 38.8 | 45.8 |
| TSDAE | 29.3 | 62.8 | 76.1 | 31.8 | 39.4 | 47.9 |
| Condenser | 25.0 | 61.7 | 73.2 | 33.4 | 41.1 | 46.9 |
| COCO-DR | 30.7 | 70.9 | **80.7** | **37.0** | 44.3 | 52.7 |
| **ABEL** | 31.1 | **73.5** | 72.7 | 35.0 | 48.4 | 52.1 |
| **ABEL-FT** | **34.3** | 72.6 | 76.5 | 36.9 | **50.0** | **54.1** |

Table 6: The comparison of unsupervised pre-training methods on representative BEIR tasks following Wang et al. (2022). The best and second-best results are marked in **bold** and underlined. The tasks are FQ=FiQA, CF=SciFact, TC=TREC-COVID, CQ=CQADupStack, RB=Robust04. Note that ABEL has not used MS-MARCO's training data for fine-tuning.

## B Effects of Noise Injection and Soft Labels

We find that injecting noise into the inputs for model training leads to better performance. Specifically, we corrupt the query and passage texts by sequentially applying random shuffling, deletion, and masking on 10% of the words. Table 4 shows the results of injecting noise into the inputs during the training of the dense retriever and the reranker, along with the use of soft labels for reranker training. We observe that noise injection results in positive effects on both retriever and reranekr training. Furthermore, incorporating soft labels in reranker training leads to additional benefits.

## C Comparison of Unsupervised Pre-training Methods

We also compare ABEL with a range of unsupervised domain-adaption methods that employ pre-training on the target corpus, SimCSE, ICT (Lee et al., 2019), MLM (Devlin et al., 2019), TS-DAE (Wang et al., 2021), Condenser (Gao and Callan, 2021) and COCO-DR. These methods follow a two-stage training paradigm, which first pre-trains a dense retriever on the target corpus (i.e., BEIR) and then fine-tunes the model using MS-MARCO training data. As shown in Table 6, our ABEL model, which solely uses unlabelled text corpus for unsupervised training, already outperforms 5 out of 6 models without requiring any supervised fine-tuning. Building on ABEL, ABEL-FT achieves the best results among all models that adopt the two-stage training paradigm. These results clearly demonstrate the superiority of our method compared to existing ones that rely on the target corpus for unsupervised pre-training.

## D Reranking Performance

We show the reranking results of ABEL-Rerank on different subsets of BEIR, comparing them with various state-of-the-art rerankers. Table 5 demonstrates that ABEL-Rerank achieves comparable or superior performance than both supervised and unsupervised rerankers, using similar model sizes across all subsets. In addition, Table 7 presents the detailed results of ABEL-Rerank (110M) for different iterations. The reranker models trained with

| ABEL-Rerank | FQ | SF | AA | CF | DB | CQ | QU | SD | FE | NF | TC | T2 | HP | NQ | RB | TN | SG | BA | **Avg.** |
|---|---|---|---|---|---|---|---|---|---|---|---|---|---|---|---|---|---|---|---|
| $t = 1$ | 34.7 | **73.4** | 55.2 | 22.7 | 37.0 | 36.1 | 82.8 | 16.5 | 68.5 | 34.4 | 74.3 | 27.9 | **67.5** | 42.0 | 49.4 | 45.6 | **31.5** | 48.6 | 47.1 |
| $t = 2$ | **35.2** | 73.1 | **55.8** | 25.2 | 37.5 | **36.8** | **82.9** | 17.1 | 76.1 | **34.8** | **77.3** | 29.5 | 66.6 | **43.3** | **50.2** | **47.1** | 31.0 | 47.9 | 48.2 |
| $t = 3$ | 35.0 | 72.9 | 53.4 | **26.7** | **37.7** | 36.6 | 82.6 | **17.1** | **80.8** | 34.2 | 76.4 | **32.1** | 67.2 | 42.7 | 49.7 | 45.4 | 29.3 | **49.0** | **48.3** |

Table 7: The comparison of bootstrapping on the reranking performance of individual tasks. The tasks are FQ=FiQA, SF=SciFact, AA=ArguAna, CF=Climate-FEVER, DB=DBPedia, CQ=CQADupStack, QU=Quora, SD=SCIDOCS, FE=FEVER, NF=NFCorpus, TC=TREC-COVID, T2=Touché-2020, HP=HotpotQA, NQ=Natural Questions, RB=Robust04, TN=TREC-NEWS, SG=Signal-1M, BA=BioASQ.

| Domain | Dataset | Query | Passage |
|---|---|---|---|
| Wikipedia | AmbigQA | Question | Question |
| Wikipedia | WikiQA | Question | Answer Sentence |
| Technical | GooAQ-Technical | Question | StackOverflow Answer |
| Technical | LinkSO-Python | Question | StackOverflow Question |
| Code | CodeSearchNet-Python | Comment | Python Code |
| Wikipedia | Natural Questions | Question | Answer Paragraph |
| Wikipedia | TriviaQA | Question | Answer Paragraph |
| Wikipedia | WebQuestions | Question | Answer Paragraph |
| Wikipedia | SQuAD | Question | Answer Paragraph |
| Wikipedia | EntityQuestions | Question | Answer Paragraph |

Table 8: Domains and task formats of cross-task evaluation datasets.

| | AMB | WQA | GAT | LSO | CSP | Avg. |
|---|---|---|---|---|---|---|
| BM25+CE | 96.8 | 84.3 | 74.7 | **32.7** | 34.9 | 64.7 |
| Contriever+CE | 96.9 | **87.0** | 74.7 | 29.1 | 43.5 | 66.2 |
| BM25+MonoT5 | 92.9 | 86.2 | 79.9 | 27.8 | 34.8 | 64.3 |
| TART | 91.1 | 82.1 | **80.5** | 25.1 | **51.4** | 66.0 |
| **ABEL-Rerank** | **97.3** | 79.8 | 77.7 | 31.4 | 45.9 | **66.4** |

Table 9: The cross-task reranking result comparison of state-of-the-art rerankers with ABEL-Rerank (nDCG@10). The best and second-best results are marked in **bold** and underlined. AMB=AmbigQA, WQA=WikiQA, GAT=GooAQ-Technical, LSO=LinkSO-Python, CSP=CodeSearchNet-Python.

more than one iteration (i.e., $t = 2, 3$) improve the performance significantly. ABEL-Rerank ($t = 2$) performs the best on more tasks (10/18), while ABEL-Rerank ($t = 3$) achieves a marginally higher overall score.

## E Cross-Task Evaluation

To validate the generalisation ability of ABEL, we conduct evaluation on a wide range of datasets (Table 8) without any further training, including AmbigQA (Min et al., 2020), WikiQA (Yang et al., 2015), GooAQ-Technical (Khashabi et al., 2021), LinkSO-Python (Liu et al., 2018), CodeSearchNet-Python (Husain et al., 2019), and five open-domain question answering datasets, namely Natural Questions (NQ) (Kwiatkowski et al., 2019),

TriviaQA (Joshi et al., 2017), WebQuestions (WebQ) (Berant et al., 2013) , SQuAD (Rajpurkar et al., 2016) and EntityQuestions (EQ) (Sciavolino et al., 2021). Since these datasets were unseen when training ABEL, their data (i.e., the text corpus) will only be accessed during the testing phase. Consequently, we consider this as a way to evaluate the capacity of ABEL to be generalised to unseen tasks and domains.

### E.1 Cross-Task Reranking Results

We compare our ABEL-Rerank model with various supervised rerankers on five datasets that were not encountered during training. The results, as shown in Table 9, indicate that ABEL-Rerank achieves the highest performance and outperforms supervised rerankers of comparable sizes (i.e., MonoT5 and TART). This finding provides compelling evidence that the reranker component involved in our approach demonstrates the ability to generalise to unfamiliar domains and tasks, rather than simply relying on memorising the corpus of each task in the BEIR benchmark to achieve promising results.

### E.2 Cross-Dataset Results on Open-Domain Question Answering Datasets

We further evaluate ABEL on five open-domain question-answering datasets that were not encountered during training to test its cross-dataset generalisation ability. We compare ABEL with a wide

| Models | NQ | | TriviaQA | | WebQ | | SQuAD | | EQ | | Avg. | |
|---|---|---|---|---|---|---|---|---|---|---|---|---|
| | Top-20 | Top-100 | Top-20 | Top-100 | Top-20 | Top-100 | Top-20 | Top-100 | Top-20 | Top-100 | Top-20 | Top-100 |
| BM25 | 63.0 | 78.2 | 76.4 | 83.1 | 62.3 | 75.5 | 71.1 | 81.8 | 71.4 | 80.0 | 68.8 | 79.7 |
| SimCSE | 43.6 | 58.5 | 62.1 | 74.3 | 44.8 | 57.7 | 47.6 | 63.3 | 38.5 | 53.9 | 47.3 | 61.5 |
| ICT† | 50.6 | 66.8 | 57.5 | 73.6 | 43.4 | 65.7 | 45.1 | 65.2 | - | - | - | - |
| MSS† | 59.8 | 74.9 | 68.2 | 79.4 | 49.2 | 68.4 | 51.3 | 68.4 | - | - | - | - |
| REALM | 61.4 | 74.8 | 72.2 | 80.1 | 62.4 | 74.1 | 47.3 | 63.5 | 64.0 | 74.1 | 61.5 | 73.3 |
| Spider† | 68.3 | 81.2 | 75.8 | 83.5 | 65.9 | 79.7 | 61.0 | 76.0 | 66.3 | 77.4 | 67.5 | 79.6 |
| Contriever | 67.2 | 81.3 | 74.2 | 83.2 | 65.7 | 79.8 | 63.0 | 78.3 | 63.9 | 75.7 | 66.8 | 79.7 |
| **ABEL** | **74.7** | **85.0** | **80.5** | **85.7** | **73.6** | **83.4** | **74.7** | **84.8** | **72.1** | **80.3** | **75.1** | **83.8** |

Table 10: Zero-shot cross-dataset retrieval results of unsupervised models on open-domain question answering datasets. Top-20 & Top-100 retrieval accuracy on the test set of each dataset is reported. The best and second-best results are marked in **bold** and underlined. Unavailable results are denoted with -. Results reported by Ram et al. (2022) are denoted with †.

| | FQ | SF | AA | CF | DB | CQ | QU | SD | FE | NF | TC | T2 | HP | NQ | RB | TN | SG | BA | QS | SR | Avg. |
|---|---|---|---|---|---|---|---|---|---|---|---|---|---|---|---|---|---|---|---|---|---|
| ABEL | 31.1 | 73.5 | 50.5 | 25.3 | 37.5 | 35.0 | 83.9 | 17.5 | 74.1 | 33.8 | 72.7 | 29.5 | 61.9 | 42.0 | 48.4 | 48.0 | 30.8 | 41.3 | 44.2 | 48.7 | 46.5 |
| ANCE | 29.5 | 51.0 | 41.9 | 20.0 | 28.1 | 29.8 | 85.2 | 12.2 | 66.7 | 23.5 | 64.9 | 24.0 | 45.1 | 44.4 | 39.2 | 38.4 | 24.9 | 30.9 | 34.0 | 43.8 | 38.9 |
| + ABEL | 34.1 | 74.1 | 51.3 | 27.4 | 40.5 | 37.3 | 87.2 | 17.9 | 79.8 | 33.9 | 76.5 | 28.6 | 64.7 | 49.5 | 50.6 | 49.3 | 30.6 | 46.1 | 47.2 | 50.6 | 48.9 |
| Contriever | 32.9 | 67.7 | 44.6 | 23.7 | 41.3 | 34.5 | 86.5 | 16.5 | 75.8 | 32.8 | 59.6 | 23.0 | 63.8 | 49.5 | 47.6 | 42.8 | 19.9 | 38.3 | 43.2 | 45.8 | 44.5 |
| + ABEL | 31.7 | 73.8 | 50.6 | 25.6 | 38.8 | 35.4 | 84.5 | 17.6 | 75.0 | 34.0 | 73.2 | 29.2 | 62.9 | 43.1 | 48.8 | 48.1 | 30.9 | 42.2 | 44.9 | 48.9 | 46.9 |
| RetroMAE | 31.6 | 65.3 | 43.3 | 23.2 | 39.0 | 34.7 | 84.7 | 15.0 | 77.4 | 30.8 | 77.2 | 23.7 | 63.5 | 51.8 | 44.7 | 42.8 | 26.5 | 42.1 | 45.8 | 45.2 | 45.4 |
| + ABEL | 34.5 | 71.4 | 48.5 | 26.8 | 43.8 | 36.3 | 86.7 | 17.5 | 82.0 | 34.2 | 80.5 | 28.8 | 69.1 | 53.8 | 51.5 | 47.2 | 30.7 | 47.9 | 49.3 | 49.7 | 49.5 |
| DRAGON+ | 35.6 | 67.9 | 46.9 | 22.7 | 41.7 | 35.4 | 87.5 | 15.9 | 78.1 | 33.9 | 75.9 | 26.3 | 66.2 | 53.7 | 47.9 | 44.4 | 30.1 | 43.3 | 47.2 | 47.6 | 47.4 |
| + ABEL | 35.5 | 72.3 | 51.3 | 25.4 | 43.6 | 37.5 | 87.8 | 18.2 | 80.2 | 34.8 | 79.0 | 28.8 | 68.5 | 53.7 | 51.2 | 48.4 | 31.4 | 46.9 | 49.1 | 50.3 | 49.7 |
| ABEL+ | 34.3 | 72.6 | 56.9 | 21.8 | 41.4 | 36.9 | 84.5 | 17.4 | 74.1 | 35.1 | 76.5 | 19.5 | 65.7 | 50.2 | 50.0 | 45.4 | 28.0 | 45.4 | 46.5 | 48.5 | 47.5 |
| + ABEL | 33.8 | 73.9 | 54.5 | 24.7 | 41.8 | 37.2 | 85.3 | 18.0 | 77.2 | 34.9 | 77.4 | 24.0 | 66.3 | 49.5 | 50.1 | 48.1 | 29.9 | 45.9 | 47.1 | 49.9 | 48.5 |

Table 11: The comparison of various supervised dense retrievers combining ABEL on BEIR benchmark using nDCG@10. QS and SR means the average performance on query-search and semantic-relatedness tasks, respectively. The tasks are FQ=FiQA, SF=SciFact, AA=ArguAna, CF=Climate-FEVER, DB=DBPedia, CQ=CQADupStack, QU=Quora, SD=SCIDOCS, FE=FEVER, NF=NFCorpus, TC=TREC-COVID, T2=Touché-2020, HP=HotpotQA, NQ=Natural Questions, RB=Robust04, TN=TREC-NEWS, SG=Signal-1M, BA=BioASQ.

range of unsupervised retrievers, including BM25, REALM, SimCSE, ICT, MSS (Sachan et al., 2021), Spider (Ram et al., 2022) and Contriever. We use Top-$k$ ($k = 20, 100$) as the main metrics for evaluation according to Karpukhin et al. (2020). As shown in Table 10, ABEL significantly outperforms other unsupervised retrievers that have been sophisticatedly pre-trained, with an average accuracy improvement of +8.4% for Top-20 and +4.0% for Top-100 over Contriever. Note that ABEL exhibits promising results and surpasses BM25 on SQuAD, a dataset with high lexical overlaps between questions and answer paragraphs, and EQ, a dataset consisting of entity-centric queries. This further confirms that ABEL effectively retains and enhances the lexical-matching ability acquired through learning from BM25 and this strength generalises well to unseen datasets.

## F Combination with Supervised Models

Table 11 shows the results in each individual task when combining ABEL with supervised dense retrievers. The findings indicate that on both types of tasks (i.e., query-search and semantic-relatedness), the ensembling with ABEL leads to performance improvements on all supervised retrievers, with the gains being generally more significant on semantic-relatedness tasks. This demonstrates the complementarity between ABEL and existing supervised dense retrievers, which are commonly trained using labelled data in the form of natural questions.

## G Effects of Boostrapping

Figure 6 shows the detailed results of each individual task throughout the iterative alternating distillation process. We observe that ABEL-1 outperforms ABEL-0 on almost all datasets, and the performance consistently improves from ABEL-1 to ABEL-2. For

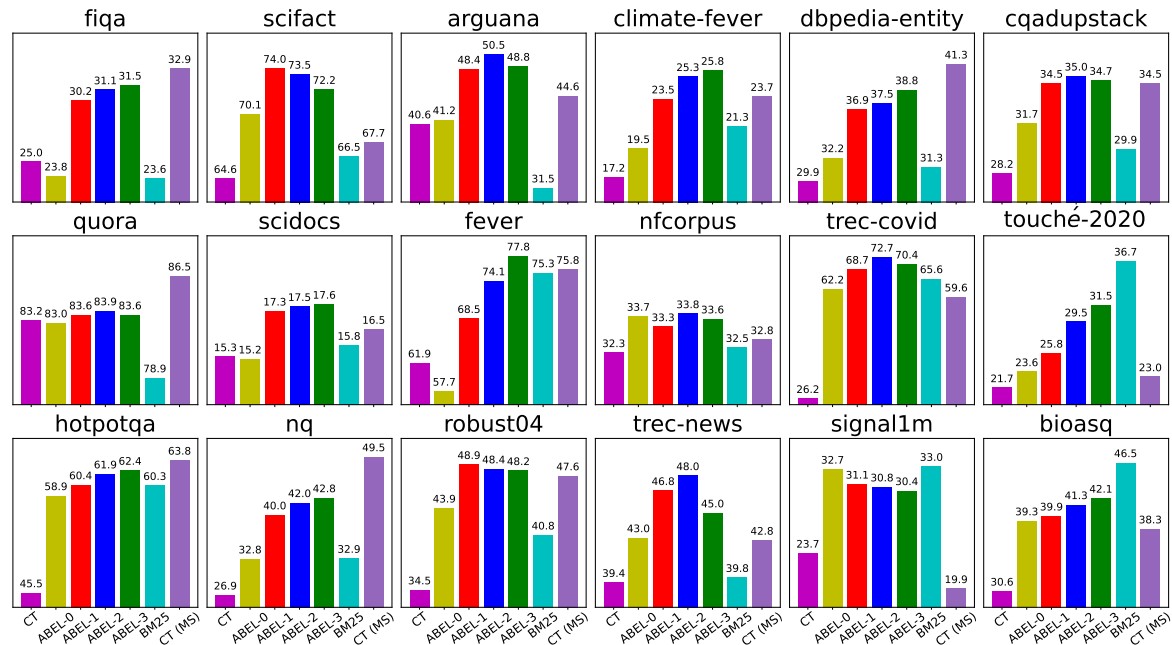

Figure 6: Effects of bootstrapping the retrieval ability on each task of BEIR. CT and MS indicate Contriever and MS-MARCO, respectively.

ABEL-3, it achieves improvement on 9 datasets but the results degrade on 6 datasets, with the overall performance merely competitive to ABEL-2.

## H The Effects of Synthetic Queries

Figure 7 shows that synthetic queries demonstrate greater efficacy on tasks involving natural questions (e.g., FiQA), while cropping-sentence queries are more effective on semantic-relatedness tasks with short sentence queries (e.g., Climate-FEVER). Furthermore, when considering query-search tasks whose domains are substantially distinct from MS-MARCO, where the query generator is fine-tuned, employing synthetic queries leads to significantly worse results, such as touché-2020 and BioASQ.

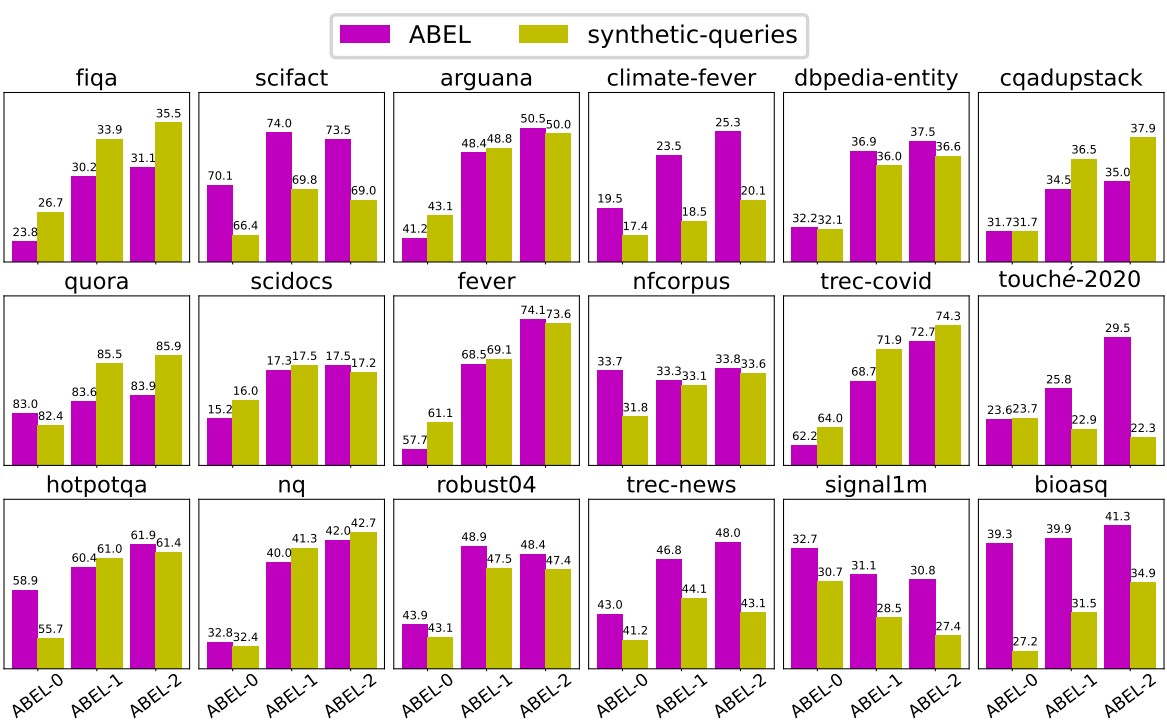

Figure 7: The comparison of cropping-sentence queries and synthetic queries on each task.

| Motivation | | | |
|---|---|---|---|
| *Practical* | *Cognitive* | *Intrinsic* | *Fairness* |
| ☐ | | | |

| Generalisation type | | | | | |
|---|---|---|---|---|---|
| *Compositional* | *Structural* | *Cross Task* | *Cross Language* | *Cross Domain* | *Robustness* |
| | | ☐ | | ☐ | |

| Shift type | | | |
|---|---|---|---|
| *Covariate* | *Label* | *Full* | *Assumed* |
| | | | ☐ |

| Shift source | | | |
|---|---|---|---|
| *Naturally occuring* | *Partitioned natural* | *Generated shift* | *Fully generated* |
| ☐ | | | |

| Shift locus | | | |
|---|---|---|---|
| *Train–test* | *Finetune train–test* | *Pretrain–train* | *Pretrain–test* |
| | | | ☐ |