# OpenReview forum: "Boot and Switch: Alternating Distillation for Zero-Shot Dense Retrieval"
_EMNLP/2023/Conference — EMNLP 2023 Findings_

### Official Review · Reviewer_ACNC · 2023-07-26

**Soundness:** 2

**Excitement:**

3: Ambivalent: It has merits (e.g., it reports state-of-the-art results, the idea is nice), but there are key weaknesses (e.g., it describes incremental work), and it can significantly benefit from another round of revision. However, I won't object to accepting it if my co-reviewers champion it.

**Paper Topic And Main Contributions:**

The authors propose ABEL, a simple but effective unsupervised method to enhance passage retrieval in zero-shot settings. Among them, the core contribution lies in the boot and switch of retriever and reranker in the iterative process. Specifically, the paper follows a straightforward loop: a dense retriever learns from supervision signals provided by a reranker, and subsequently, the reranker is updated based on feedback from the improved retriever. By iterating this loop, the two components mutually enhance one another’s performance.

**Questions For The Authors:**

1.	In general, bootstrapping requires some variation in the source or presentation of information, and then improved competence through self-learning. In this paper, how is the superiority of labeled data ensured during the iterative process? In addition, what are the specific trends of change between what the retriever and reranker learn and what can be instructed to each other in the process of learning data?
2.	“their labels can be iteratively refined by the retriever and the reranker to enhance the training of the other model.” The approach of alternating improved labels creates dependencies between the retriever and the reranker, and how the authors consider the magnitude of the ability to process the data between the two to prevent the emergence of some pernicious dependencies (singly optimized labels) as well as the disappearance of benign dependencies (co-optimized labels).
3.	The authors observe that using soft labels captures subtle semantic relationships between multiple related passages and produces better results than using hard labels to train the reranker. Does the above arise from inaccuracies in hard labels information? If so, which information produces better results in the Zero-shot condition when the semantic information obtained by hard labels is closer to the real sample attributes compared to the relationship information obtained by soft labels? If not, can we understand that inter-paragraph relational information is more important than attribute information in any case?
4.	Please ensure that there are corresponding notations at the indexes in the paper, e.g. line 249 lacks a notation for equation (5).


**Reasons To Accept:**

1.	The paper is written clearly and is well-organized. Technical details are sufficient so that I am confident the results would be reproducible.
2.	Lots of experiments to support the methodology. The authors perform overall good discussions on the components of their work, showing the boost from each part. A lot of visualized results are illustrated, e.g., including the Zero-shot retrieval results on BEIR (nDCG@10), Reranking results on BEIR, the effect of bootstrapping the retrieval ability on BEIR, and so on.


**Reasons To Reject:**

1.	The novelty of this paper is not sufficient. We found that it is not uncommon for collaborative work between teachers and students, such as those represented by DML[1], RocketQAv2[2], etc. In addition, the impact of noise on model performance in this paper is also similar to the data augmentation in the above existing work.
[1]	Zhang Y, Xiang T, Hospedales T M, et al. Deep Mutual Learning [M]. CVPR. 2018: 4320-4328.
[2]	RocketQAv2: A Joint Training Method for Dense Passage Retrieval and Passage Re-ranking. EMNLP (1) 2021: 2825-2835
2.	Contributions are not well argued. The core contribution of the paper lies in the Boot and Switch between retriever and reranker. However, the paper introduces the working mechanism of the two, but rarely discusses the internal reasons for the Boot and Switch between retriever and reranker.


**Reproducibility:**

4: Could mostly reproduce the results, but there may be some variation because of sample variance or minor variations in their interpretation of the protocol or method.

**Reviewer Confidence:**

4: Quite sure. I tried to check the important points carefully. It's unlikely, though conceivable, that I missed something that should affect my ratings.

---

> ### Author Rebuttal · Authors · 2023-08-28
>
> Thanks for your comments and suggestions.
>
> ---
>
> > W1: Novelty and comparison to previous work
>
> We would like to highlight the novel aspects of our work to previous works:
>  + DML [1] jointly trains multiple networks with identical model capacity through peer distillation. Our method uses models with different roles (i.e., reranker and retriever) and enforces mutual improvement between the participants. We believe our design and corresponding research are novel regarding adapting knowledge distillation to the complex setting in IR research.
>  + RocketQAv2 [2] jointly trains the retriever and the ranker using knowledge distillation, but with the effect of a single direction, namely using the reranker to improve the retriever. By contrast, we suggest two insights (1) a model with a larger capacity can be improved by a less capable model and (2) we can iteratively realise mutual improvement between two models.
>
> We believe our proposed iterative alternating distillation algorithm is not trivial as it shows that, for the first time, two models with different modeling abilities can mutually benefit in an entirely unsupervised manner, and this can result in models that surpass sophisticated supervised retrievers. Our research is beneficial to search engines in a “cold-start” setting when no labeled data is available. The performance of ABEL can be further improved by training on accumulated user data from search logs.
>
> ---
>
> > W2: “Internal reasons for the Boot and Switch between retriever and reranker”
>
> There are two main reasons that inspire our design:
> 1.  Better rerankers potentially correct the errors made by retrievers and consequently improve label qualities. By introducing an additional more capable model as a reranker, it is able to reorder the predictions from the retriever by reranking passages that are closer to gold ones higher (more likely to be regarded as positives) and sample more contrastive negatives (relevant to but not answering the query) from its predictions.
> 2. An improved retriever is able to generate harder negative samples with more accurate soft labels to reflect the nuanced relatedness between passages and queries.
>
> An alternative distillation method is always to learn from the data generated by itself. We have verified that this paradigm underperforms our approach (see the comparison in Figure 5(a), the purple line vs the blue line), with the performance saturating after one iteration. Moreover, the retriever learned from itself quickly converges to fixed labeling behavior.  We believe this results from the shortage of error corrections, which is the role the reranker played in our method, and the diverse samples and accurate labelling introduced by our alternating learning are essential
>
> ---
>
> > Q1: “How is the superiority of labeled data ensured during the iterative process? In addition, what are the specific trends of change between what the retriever and reranker learn and what can be instructed to each other in the process of learning data?”
>
> The key intuition is that (1) a better retriever provides better candidate sets (more semantically related to queries) to train a better reranker and (2) a better reranker provides more accurate positives and more contrastive negatives to train a better retriever. The two models bring very different “views” (in the sense of co-training & multi-view learning) to the problem: one imposes a factorisation with a tight bottleneck, while the other supports full cross-attention. These different architectures affect the tractability of search, but also the capacity to model the full problem, and the propensity to overfit.
>
> Empirically, we found that the performance of both retriever and reranker keeps improving during the training iterations. This indicates a promising direction that the generated labels by both models are converging to the distributions with higher quality.
>
> ---
>
> > Q2: “How the authors consider the magnitude of the ability to process the data between the two to prevent the emergence of some pernicious dependencies (singly optimized labels) as well as the disappearance of benign dependencies (co-optimized labels)?”
>
> We can interpret the retriever as a proposal distribution generator, which selects relatively good candidates, and the reranker as a reward scoring model, which measures the quality of an example as a good answer. They can work cooperatively to generate labels of higher quality.
>
> At each iteration, the improved reranker can be used to correct the predictions from the retriever, therefore reducing the number of inaccurate labels in retriever training data. Meanwhile, the improved retriever is more likely to include more relevant passages in its top-k predictions, which increases the chance of finding correct labels by the reranker. Overall,  as training goes on, (1) the range where correct answers appear can be narrowed down by the retriever (i.e. better proposal distributions) and (2) the accuracy of finding correct answers from such ranges can be increased by the reranker.
>
> ---
>
> > Q3: “Does the inaccuracies in hard label information leads to inferior result of reranker compared to soft labels? If so, which information produces better results in the Zero-shot condition when the semantic information obtained by hard labels is closer to the real sample attributes compared to the relationship information obtained by soft labels? If not, can we understand that inter-paragraph relational information is more important than attribute information in any case?”
>
> We show that using hard labels for reranker training is inferior to soft labels in Table 3 in Appendix B. Additionally, we found that the reranker trained on hard labels tends to converge to trivial solutions (i.e., predicting merely the passage where the crop-sentence query appears). Soft labels provide the reranker with more diverse training signals by learning from a distribution. Additionally, soft label is a quite natural choice, as soft labels were shown more effective than hard labels for distillation in [1].
>
> ---
>
> > Q4:  Presentation
>
> We will correct the presentation issues in our modification.
>
> ---
>
> [1] Mary Phuong and Christoph Lampert. 2019. Towards understanding knowledge distillation. In Proceedings of the 36th International Conference on Machine Learning

---

### Official Review · Reviewer_3ATb · 2023-08-03

**Soundness:** 4

**Excitement:**

4: Strong: This paper deepens the understanding of some phenomenon or lowers the barriers to an existing research direction.

**Paper Topic And Main Contributions:**

The paper proposes to a training strategy to iteratively train retriever and reranker. The authors demonstrate that starting from unsupervised labels using BM25, both the retriever and reranker can reach competitive even better effectiveness than many supervisedly trained rankers.

**Questions For The Authors:**

1. Have you tried using the same training data as DRAGON[1] (i.e., MS MARCO cropped sentences)? In this way, we can see a fair comparison (in terms of data) between the proposed approaches of ABEL and DRAGON.  Also, this can make reader understand how much gain from data augmentation on diverse corpus in BEIR rather than single corpus in MS MARCO.
2. The authors have found that the sentence-cropped trained models are weaker in query-search tasks; then, why the authors choose not to include synthetic queries in the training process? Does it harms the overall performance?

[1] Sheng-Chieh Lin, Akari Asai, Minghan Li, Barlas Oguz, Jimmy Lin, Yashar Mehdad, Wen-tau Yih, and Xilun Chen. 2023a. How to Train Your DRAGON: Diverse Augmentation Towards Generalizable Dense Retrieval. ArXiv:2302.07452



**Reasons To Accept:**

1. The paper introduces an approach to train retriever and reranker in a completely unsupervised manner and reach best ranking effectiveness among all unsupervised models.
2. The experiments are comprehensive, including the comparisons on cross tasks with unsupervised models, the comparisons of different training strategies. The experiments clearly illustrates the importances of introducing cross-encoder in the alternating distillation and cropped sentences as training queries.

**Reasons To Reject:**

1. The proposed approach is basically the combination of the existing work. I would consider the main contribution of the paper three folds: cheap and large-scale data augmentation, diverse domain, alternating distillation with retriever and reranker. However, these two approaches are not new. For example, this sentence cropped approach has introduced by Chen et al[1]; Using training data from divers domain is introduced by Ye et al[2]; alternating distillation is also introduced by the previous work[3].


[1] Xilun Chen, Kushal Lakhotia, Barlas Oguz, Anchit Gupta, Patrick Lewis, Stan Peshterliev, Yashar Mehdad, Sonal Gupta, and Wen tau Yih. 2021. Salient phrase aware dense retrieval: Can a dense retriever imitate a sparse one? In Proc. EMNLP.
[2] Yue Yu, Chenyan Xiong, Si Sun, Chao Zhang, and Arnold Overwijk. 2022. Coco-dr: Combating distribution shifts in zero-shot dense retrieval with contrastive and distributionally robust learning. In Proc. EMNLP.
[3] Ruiyang Ren, Yingqi Qu, Jing Liu, Wayne Xin Zhao, QiaoQiao She, Hua Wu, Haifeng Wang, and Ji-Rong Wen. 2021. RocketQAv2: A joint training method for dense passage retrieval and passage re-ranking. In Proc. EMNLP, pages 2825–2835.


**Reproducibility:**

4: Could mostly reproduce the results, but there may be some variation because of sample variance or minor variations in their interpretation of the protocol or method.

**Reviewer Confidence:**

4: Quite sure. I tried to check the important points carefully. It's unlikely, though conceivable, that I missed something that should affect my ratings.

---

> ### Author Rebuttal · Authors · 2023-08-28
>
> Thanks for your summarization of our contributions and suggestions.
>
> ---
>
> > W1: The Difference from Previous Methods
>
> Thanks for acknowledging our contributions on the unsupervised iterative alternating distillation methods. We would like to clarify our novelty to [1], [2], and [3]:
> 1. Chen et al [1] mainly focus on how to best distill the knowledge from sparse retrievers, while our work considers crop sentences as a primary source of obtaining unsupervised training signals cheaply.
> 2. Yu et al [2] primarily take the diverse corpus from BEIR for model pre-training to improve the domain-adaption ability, while our work shows that the model can outperform a wide range of supervised ones without using supervised data.
> 3. Ren et al [3] jointly train a retriever and a reranker through knowledge distillation and self-mined pseudo positives. However, their work doesn’t explore the alternating distillation mechanism, which is introduced in Section 3.2 and verified in Figure 4. The significance of our work lies in that we are the first to show the effectiveness of (bi-directional) alternating distillation for unsupervised IR, while [3] only discusses one-directional knowledge flow (using reranker to improve retriever), without the reversed effect (using retriever to improve reranker).
>
> ---
>
> > Q1: “Have you tried using the same training data as DRAGON[1] (i.e., MS MARCO cropped sentences)?”
>
> In our preliminary experiments, we have tried using only cropped sentences from MS MARCO as the training data. This method on BEIR is significantly better than the vanilla Contriever (Avg. 41.0 vs Avg. 35.9) but is inferior to the one using diverse corpus from BEIR, and we believe that the diverse corpora we used in training is one of the factors to our success.
>
> ---
>
> > Q2: “Why the authors choose not to include synthetic queries in the training process?”
>
> There are two main reasons for not including synthetic queries:
> 1. It incurs additional cost, especially when using large language models to generate high-quality synthetic queries.
> 2. In our experiments, we did not observe the advantage of using synthetic queries, compared with those using crop sentences in both query-search and semantic-relatedness tasks, as shown in Figure 5(a). Please also refer to the corresponding comparison in Figure 7 in Appendix H.
>
> ---
>
> [1] Xilun Chen, Kushal Lakhotia, Barlas Oguz, Anchit Gupta, Patrick Lewis, Stan Peshterliev, Yashar Mehdad, Sonal Gupta, and Wen-tau Yih. 2022. Salient phrase aware dense retrieval: Can a dense retriever imitate a sparse one? In Findings of the Association for Computational Linguistics: EMNLP 2022
>
> [2] Yue Yu, Chenyan Xiong, Si Sun, Chao Zhang, and Arnold Overwijk. 2022. COCO-DR: Combating the distribution shift in zero-shot dense retrieval with contrastive and distributionally robust learning. In Proceedings of the 2022 Conference on Empirical Methods in Natural Language Processing
>
> [3] Ruiyang Ren, Yingqi Qu, Jing Liu, Wayne Xin Zhao, QiaoQiao She, Hua Wu, Haifeng Wang, and Ji-Rong Wen. 2021. RocketQAv2: A joint training method for dense passage retrieval and passage re-ranking. Proceedings of the 2021 Conference on Empirical Methods in Natural Language Processing

---

### Official Review · Reviewer_jdQL · 2023-08-03

**Typos Grammar Style And Presentation Improvements:** 1. The definition of $P$ and $P^k_D(q…
**Soundness:** 3

**Excitement:**

3: Ambivalent: It has merits (e.g., it reports state-of-the-art results, the idea is nice), but there are key weaknesses (e.g., it describes incremental work), and it can significantly benefit from another round of revision. However, I won't object to accepting it if my co-reviewers champion it.

**Paper Topic And Main Contributions:**

This paper proposes ABEL, an Alternating Bootstrapping training framework for unsupervised dense rEtrievaL. The method alternates the distillation process between a dense retriever and a reranker by switching their roles as teachers and students in iterations. Contriever is used to initialize the dense retriever, and the reranker is initialized from t5-base-lm-adapt. The method starts with warming up the retriever by imitating BM25, then (1) training a reranker based on the labels extracted from the retriever by the last iteration; (2) refining the dense retriever using training signals derived from the reranker by the last step. Experimental results show that ABEL achieves superior performance in the unsupervised setting when compared with BM25 and Contriever.

**Questions For The Authors:**

1. Does the term "zero-shot" in the title refer to the dataset having no manual labels (unsupervised), the dataset being constructed without reliance on synthetic queries, or the ability to perform cross-domain inference?

2. Have the authors considered any methods to detect and avoid error accumulation in the loop of model training?

**Reasons To Accept:**

1. The framework proposed in this paper is relatively simple and practical.

2. The experiment is extensive.

**Reasons To Reject:**

1. The overall structure of the article is clear, but some details are confusing. For example, the definition of zero-Shot dense retrieval, their differences and connections among ABEL, ABEL+, and ABEL++.

2. The baselines seems to be weak. Only BM25 and Contriever are involved in the unsupervised setting. And under the supervised setting, the improvement of ABEL+ over SPLADE++ and DRAGON+ seems marginal.

**Reproducibility:**

4: Could mostly reproduce the results, but there may be some variation because of sample variance or minor variations in their interpretation of the protocol or method.

**Reviewer Confidence:**

2: Willing to defend my evaluation, but it is fairly likely that I missed some details, didn't understand some central points, or can't be sure about the novelty of the work.

---

> ### Author Rebuttal · Authors · 2023-08-28
>
> Thanks for your comments and suggestions!
>
> ---
>
> > W1: Differences among ABEL, ABEL+ and ABEL++
>
> A: (1) ABEL+ is the model obtained by further fine-tuning ABEL on the MS-MARCO dataset and (2) ABEL++ adds a reranker to rerank the top-k predictions from ABEL, as described in Section 4.2. We will rename ABEL+ and ABEL++ to ABLE+FT (fine-tuning) and ABEL+FT+reranker respectively to highlight their difference and modify our description accordingly.
>
> ---
>
> > W2 (a): “The baselines seems to be weak. Only BM25 and Contriever are involved in the unsupervised setting.”
>
> A: (1) We include BM25 [1] and Contriever [2] as baselines in Table 1 to represent competitive lexical-based and neural-based methods, respectively. For some retrieval tasks, these baselines have been proven very competitive, and BM25 can outperform the best supervised neural models [3].  (2) We have included four unsupervised approaches, i.e., BM25, REALM, SimCSE, and Contriever, as compared in Table 2. The two selected baselines used in Table 1 have the highest overall performance. (3) (Unsupervised) ABEL outperforms all these baselines as demonstrated in Tables 1 and 2. We will clarify our selection process for the unsupervised baselines in our revision.
>
> > W2 (b): Marginal Improvements over SPALDE++ and DRAGON+
>
> [Compared with the best lexical model (SPLADE++)]: Although lexical models generally outperform dense models as they map learned vectors to actual lexicons, our dense model (ABEL+) (1) is more efficient in scaling to massive documents by using GPUs; (2) manage to outperform the best lexical-based approach.
>
> [Compared with DRAGON+, a dense model learning from SPALDE++ and other models] Although DRAGON+ enjoys two advantages: (1) inheriting knowledge from a lexical-based model and (2) using a more advanced pre-trained model (RetroMAE) as backbone than ours (Contriever), our method can still outperform this baseline (DRAGON+). To have a fair comparison, we add experimental results of both methods using aligned pre-trained models as follows:
>
>  | Initialisation | BERT | Contriever | RetroMAE |
> | :--: | :----: | :------: | :----: |
> | DRAGON+ | 46.8 | N/A | 47.4 |
> | ABEL+ | 47.2 | 47.5 | 48.0 |
> |Improv. | +0.4 | N/A | +0.6 |
>
> The improvement by our approach is consistent, higher than 0.4%. We choose Contriever as our base model to align with the settings of previous papers, i.e., Contriever[2] and PTR[4]. We will add corresponding discussions in our modification.
>
> ---
>
> > Q1: Definition of zero-shot setting.
>
> When training a retriever, the ground-truth annotations for each task are unknown - i.e., all we have are the passages, but we have no queries, nor relevance judgments.  Thus our setting is zero-shot in terms of the target task, and we assess the ability to transfer across domains. As shown in Table 2, ABEL can achieve consistent performance gains over Contriever and surpass other unsupervised baselines on unseen tasks without further training.
>
> ---
>
> > Q2: Error Accumulation
>
> We use re-initialisation to avoid the accumulation from biased errors in early iterations. At the start of each iteration, we re-initialise both the retriever and reranker using pre-trained language models rather than continuously fine-tuning the model obtained from the last iteration (more details in lines 236-241). We will highlight the setting in our revision.
>
> ---
>
> [1] Stephen Robertson and Hugo Zaragoza. 2009. The probabilistic relevance framework: Bm25 and beyond. Found. Trends Inf. Retr.
>
> [2] Gautier Izacard, Mathilde Caron, Lucas Hosseini, Sebastian Riedel, Piotr Bojanowski, Armand Joulin, and Edouard Grave. 2022. Unsupervised dense information retrieval with contrastive learning. Transactions on Machine Learning Research.
>
> [3] Nandan Thakur, Nils Reimers, Andreas Rücklé, Abhishek Srivastava, and Iryna Gurevych. 2021. BEIR: A heterogeneous benchmark for zero-shot evaluation of information retrieval models. In Thirty-fifth Conference on Neural Information Processing Systems Datasets and Benchmarks Track.
>
> [4] Zhuyun Dai, Vincent Y Zhao, Ji Ma, Yi Luan, Jianmo Ni, Jing Lu, Anton Bakalov, Kelvin Guu, Keith Hall, and Ming-Wei Chang. 2023. Promptagator: Few-shot dense retrieval from 8 examples. In The Eleventh International Conference on Learning Representations.

---

### Meta-Review · Area_Chair_9d2K · 2023-09-27

**Recommendation:** 2

**Metareview:**

This paper proposes ABEL, a training strategy to iteratively train retriever and reranker for unsupervised dense retrieval in the zero-shot setting. The reviewers' opinions on this work are not unanimously positive. While the authors have addressed certain concerns raised by the reviewers, there are still some lingering apprehensions. The authors have provided appropriate responses to jdQL's concerns regarding method details and comparison with baselines, which resulted in an increased score from R1. Both 3ATb and ACNC expressed that the novelty and contributions of the proposed method were limited. The authors have provided detailed responses highlighting the differences between ABEL and previous methods, while also clarifying the contributions of ABEL. However, these concerns still persist.

---

### Decision · Program_Chairs · 2023-10-07

**Decision:**

Accept-Findings

**Comment:**

This paper proposes ABEL, a training strategy to iteratively train retriever and reranker for unsupervised dense retrieval in the zero-shot setting. The reviewers' opinions on this work are not unanimously positive. While the authors have addressed certain concerns raised by the reviewers, there are still some lingering apprehensions. The authors have provided appropriate responses to jdQL's concerns regarding method details and comparison with baselines, which resulted in an increased score from R1. Both 3ATb and ACNC expressed that the novelty and contributions of the proposed method were limited. The authors have provided detailed responses highlighting the differences between ABEL and previous methods, while also clarifying the contributions of ABEL. However, these concerns still persist.